# Copy Number Analysis of 9p24.1 in Classic Hodgkin Lymphoma Arising in Immune Deficiency/Dysregulation

**DOI:** 10.3390/cancers16071298

**Published:** 2024-03-27

**Authors:** Kumiko Ohsawa, Shuji Momose, Asami Nishikori, Midori Filiz Nishimura, Yuka Gion, Keisuke Sawada, Morihiro Higashi, Michihide Tokuhira, Jun-ichi Tamaru, Yasuharu Sato

**Affiliations:** 1Department of Molecular Hematopathology, Okayama University Graduate School of Health Sciences, Okayama 700-8558, Japan; oskumiko@saitama-med.ac.jp (K.O.); asami.kei331@okayama-u.ac.jp (A.N.); midorifiliz-nishimura@okayama-u.ac.jp (M.F.N.); 2Department of Pathology, Saitama Medical Center, Saitama Medical University, Saitama 350-8550, Japan; momose@saitama-med.ac.jp (S.M.); k_sawada@saitama-med.ac.jp (K.S.); mhigashi@saitama-med.ac.jp (M.H.); jtamaru@saitama-med.ac.jp (J.-i.T.); 3Department of Medical Technology, Faculty of Health Sciences, Ehime Prefectural University of Health Sciences, Tobe 791-2101, Japan; ygion@epu.ac.jp; 4Department of Hematology, Japan Community Health Care Organization Saitama Medical Center, Saitama 330-0074, Japan; 5PCL Japan, Pathology and Cytology Center, Saitama 331-9530, Japan

**Keywords:** classic Hodgkin lymphoma, methotrexate, immunodeficiency, programmed cell death-ligand 1, rheumatoid arthritis

## Abstract

**Simple Summary:**

The clinical course of classic Hodgkin lymphoma arising in immune deficiency/dysregulation (CHL-IDD) differs from that of CHL arising in immunocompetent cases and typically exhibits aggressive clinical behaviors. This study investigated the genetic aberration of 9p24.1 and protein expression of PD-L1 and also analyzed the clinicopathological association of these genetic lesions in CHL-IDD. Our findings showed that PD-L1 expression and 9p24.1 copy number alterations were observed in all patients analyzed in this study. Though it is recognized that an increase in 9p24.1 copy number alteration is associated with a more aggressive clinical course in immunocompetent CHL patients, we identified a subset of the 9p24.1 copy gain group, which had less 9p24.1 copy number alteration than the amplification group, exhibited more extensive extranodal lesions and had higher clinical stages in CHL-IDD cases. This finding speculates the presence of a genetically distinct subgroup within CHL-IDD patients, which may explain certain characteristic features.

**Abstract:**

A subset of patients with rheumatoid arthritis receiving methotrexate develop immune deficiencies and dysregulation-associated lymphoproliferative disorders. Patients with these disorders often exhibit spontaneous regression after MTX withdrawal; however, chemotherapeutic intervention is frequently required in patients with classic Hodgkin lymphoma arising in immune deficiency/dysregulation. In this study, we examined PD-L1 expression levels and 9p24.1 copy number alterations in 27 patients with classic Hodgkin lymphoma arising from immune deficiency/dysregulation. All patients demonstrated PD-L1 protein expression and harbored 9p24.1 copy number alterations on the tumor cells. When comparing clinicopathological data and associations with 9p24.1 copy number features, the copy gain group showed a significantly higher incidence of extranodal lesions and clinical stages than the amplification group. Notably, all cases in the amplification group had latency type II, while 6/8 (75%) in the copy gain group had latency type II, and 2/8 (25%) had latency type I. Thus, a subset of the copy-gain group demonstrated more extensive extranodal lesions and higher clinical stages. This finding speculates the presence of a genetically distinct subgroup within the group of patients who develop immune deficiencies and dysregulation-associated lymphoproliferative disorders, which may explain certain characteristic features.

## 1. Introduction

Immunodeficiency-associated lymphoproliferative disorders (LPDs), which can occur due to the use of immunosuppressive agents, have been proposed as other iatrogenic immunodeficiency-LPDs (OII-LPDs) in the fourth edition of the World Health Organization (WHO) classification [1]. Previous large cohorts have suggested a high incidence of lymphoma in patients with rheumatoid arthritis (RA) and other autoimmune disorders [2,3,4,5,6]. Methotrexate (MTX) is widely used as a first-line disease-modifying antirheumatic drug for the treatment of RA [7,8]. A subset of patients who receive MTX for RA develop LPDs, indicating a potential link between MTX usage and the development of LPDs [9,10,11,12,13,14]. The current fifth edition of the WHO classification changed the framework of immune deficiency and dysregulation-associated lymphoproliferative disorders (IDD-LPDs) classification from relying on the underlying background of the immunodeficiency to being based on histopathological features [15]. Of the IDD-LPDs, the most common type of lymphoma encountered is diffuse large B-cell lymphoma (DLBCL-IDD), accounting for 35–60% of cases, followed by classic Hodgkin lymphoma (CHL-IDD) at 12–28% [12,13,16]. From the clinical point of view, IDD-LPDs exhibit one unique characteristic, which is the spontaneous resolution that occurs in 25–81.8% of patients after MTX withdrawal [16,17,18,19,20,21,22,23]. While DLBCL-IDD typically exhibits spontaneous resolution without recurrence, most cases of CHL-IDD are associated with relapses that necessitate chemotherapy [5,24,25].

De novo classic Hodgkin lymphoma (de novo CHL) is characterized by histology consisting of a small fraction of tumor cells (Hodgkin/Reed–Sternberg cells) surrounded by various inflammatory cells that form the tumor microenvironment [15]. Genomic analyses have revealed structural aberrations at the 9p24.1 locus that encodes programmed cell death-ligand 1 (PD-L1) in nearly all cases of de novo CHL [15]. The high expression of PD-L1 resulting from this abnormality forms one of the molecular pathogenetic bases for the CHL tumor microenvironment and evidence for using immune checkpoint inhibitors [26]. Similarly, nearly all patients with CHL-IDD have been reported to exhibit high PD-L1 protein expression by immunohistochemistry. Moreover, a higher positivity rate is associated with a shorter time to chemotherapy initiation [25].

Roemer et al. analyzed the 9p24.1 locus by FISH in de novo CHL [27]. They reported that patients with 9p24.1 amplification had significantly shorter progression-free survival (PFS) than those with other genetic alterations, such as copy number gain or polysomy [27]. A study of the 9p24.1 region in Japanese patients with CHL-IDD detected a genetic abnormality in all cases [28]. However, only seven cases were analyzed in that study, and the relationship between the genetic finding and clinical pathophysiology was not assessed [28]. In this study, we analyzed copy number alterations (CNAs) in the 9p24.1 region, PD-L1 expression, and clinical characteristics in 27 cases of CHL-IDD.

## 2. Materials and Methods

### 2.1. Patients

We analyzed 27 patients with CHL-IDD diagnosed and treated at Okayama University, Saitama Medical University, Saitama Medical Center, and other related hospitals between 1992 and 2022. Clinical and laboratory data were extracted from medical records. The Ann Arbor classification was applied to classify the clinical stage (CS). This study was approved by the Institutional Review Board of Okayama University (Ref. No. 1607-016, 2007-033) and Saitama Medical University, Saitama Medical Center (Ref. No. 1966, 2503). Comprehensive informed consent was obtained from all patients at Okayama University and Saitama Medical University, Saitama Medical Center using the opt-out method. The cases included in this study comprised 11 cases analyzed in a previous report [25].

### 2.2. Histological Examinations

All specimens were fixed in 10% neutral-buffered formalin and sliced into three μm sections from the paraffin-embedded blocks. Sections were subjected to hematoxylin and eosin (HE) staining (Figure 1A,B), immunohistochemical staining, and fluorescence in situ hybridization (FISH) analyses. Immunohistochemical staining was performed using a Leica Bond III autoimmunostainer (Leica Biosystems, Wetzler, Germany) and a BenchMark Ultra (Ventana, Oro Valley, AZ, USA). The monoclonal antibodies used were as follows: anti-CD3 (1:100/200, LN10; Novocastra Laboratories, Ltd., Newcastle upon Tyne, UK), anti-CD5 (1:100, 4C7; Novocastra Laboratories, Ltd.), anti-CD10 (1:50/100, 56C6; Novocastra Laboratories, Ltd.), anti-CD15 (1:50, Carb-3; DAKO, Glostrup, Denmark, Leu-M 1; BD Biosciences, San Jose, CA, USA), anti-CD20 (1:100/200, L26; DAKO, Novocastra Laboratories, Ltd.), anti-CD30 (1:40, Ber-H2; DAKO, RTU, Ber-H2; Roche Diagnostics K.K., Tokyo, Japan) (Figure 1C), anti-CD79a (1:100, JCB117; DAKO), anti-Ki-67 (1:100/2500, MIB-1; DAKO), anti-PD-L1 (1:400, E1L3N; Cell Signaling Technology, Danvers, MA, USA) (Figure 1D), anti-LMP1 (1:50/100, CS1-4, Novocastra, DAKO), and anti-EBNA2 (1:100/200, PE2, Abcam, Cambridge, MA, USA). PD-L1 protein expression was scored by two pathologists (YS, SM), and the positivity of PD-L1 on tumor cells was classified into four groups: 1 for a PD-L1-positivity rate of ≤25%, 2 for 26–50%, 3 for 51–75%, and 4 for >75%. [25]. PD-L1 expression levels were also calculated using the modified H-score reported by Roemer et al. [27]. Two pathologists (YS, SM) counted 50 (or as many as possible if <50) Hodgkin/Reed Sternberg (HRS) cells and evaluated the mean intensity (0–3+) of positively stained cells (Figure 1E–H) to define the percentage of cells with mean staining intensities at each (0–100%) to generate a modified H-score.

### 2.3. Immunofluorescence In Situ Hybridization

For fluorescence immunophenotyping and interphase cytogenetics as a tool for the investigation of neoplasia (FICTION) analysis, fluorescent immunostaining for CD30 was performed using Bond III, followed by FISH analysis. Roche diluted antibody CD30; BerH2 (Roche Diagnostics K.K., Tokyo, Japan) was used as the primary antibody, and an IgG Alexa Fluor™ 488 Tyramide SuperBoost™ Kit for goat anti-mouse IgG (B40912, Thermo Fisher Scientific, Waltham, MA, USA) was used for the secondary antibody, according to the manufacturer’s protocols. The FISH probes (Figure 2A) were prepared according to a previously reported protocol [27]. The obtained FICTION images were captured and analyzed using Duet on a Bioview system equipped with an Olympus microscope (Abbott Molecular Inc., Des Plaines, IL, USA). Signals were evaluated by analyzing 50 tumor cells (or as many as possible if <50 tumor cells were present) according to the method of Roemer et al. [27]. When the ratio of PD-L1/PD-L2 to CEP9 was 1:1, it was considered representative of disomy; when the ratio was 3:1, it was considered amplification; any ratio between disomy and amplification was considered copy gain. Polysomy was defined when the ratio of PD-L1/PD-L2 to CEP9 was 1:1, but CEP was ≥ 3 (Figure 2B). Two pathologists (YS, SM) counted 50 (or as many as possible if <50) Hodgkin/Reed Sternberg (HRS) cells and chromosome 9p24.1 copy number alterations (CNAs) were evaluated according to a protocol described by Roemer et al. [27]. Cases that showed amplification were classified into the amplification group, and cases that showed no amplification but copy gain were classified into the copy-gain group according to a description by Roemer et al. [27].

### 2.4. Statistical Analyses

Associations between variables were evaluated using Fisher’s exact test for categorical data and Wilcoxon or Kruskal–Wallis rank-sum tests for continuous data across two or more groups. Survival curves were assessed using the Kaplan–Meier method and compared using log-rank tests. All statistical analyses were performed using JMP Pro 16.2.0 (JMP Statistical Discovery LLC., Cary, NC, USA).

## 3. Results

### 3.1. Patient Characteristics

The clinical data of the 27 patients with CHL-IDD are shown in Table 1 and Table 2. Of the total patients, 18 of 26 (69.2%) had CS of ≥3 or 4. Of the 27 total patients, 26 had RA, while 1 was being treated with MTX for SAPHO syndrome. All patients received MTX, but 15/27 (55.6%) also received prednisolone (PSL), and 14/27 (51.9%) were treated with antirheumatic drugs or biological agents—including sulfasalazine, bucillamine, or adalimumab. Tacrolimus was administered in 7 cases. The median observation period was 27 (0.4–151.2) months.

### 3.2. Clinical Courses after Methotrexate Withdrawal

After being diagnosed with IDD-LPDs, MTX was discontinued in 26 cases except one. In one case, MTX was discontinued for another reason, but details were not available. Of these, 16 (61.5%) experienced regression: 7 cases in complete remission (CR) and 9 cases in partial remission (PR), while 10 (38.5%) had stable disease (SD) or progressive disease (PD); Table 2, Figure 3A). In the regression group, 5/16 patients survived without requiring chemotherapy, while the remaining 11 experienced relapses or disease progression—with all but 2 receiving adriamycin, bleomycin, vinblastine, and dacarbazine (ABVD) or Brentuximab-Vedotin, adriamycin, vinblastine, and dacarbazine (A-AVD). Conversely, 9 of the patients in the SD/PD group received chemotherapy, except for 1 who declined treatment. Of the 18 patients who received chemotherapy, 12 of the patients were alive, and 6 died; the median time from MTX withdrawal to chemotherapy was 8.5 months (0.1–44.0).

### 3.3. Immunohistochemical Phenotype and Epstein–Barr Virus Latency

The results of our CNA and immunostaining analyses are presented in Table 3. CD30 was positive in all of the HRS cells in all cases. CD15 was positive in 16/22 (72.7%), and CD20 was positive in 10/27 (37.0%) patients. Epstein–Barr virus-encoded small RNA in situ hybridization (EBER-ISH) was positive in 22/27 (81.5%) cases, with 19/21 (90.5%) showing latency type II and 2/21 (9.5%) showing latency type I. No cases were positive for Epstein–Barr virus nuclear antigen 2 (EBNA-2) (i.e., latency type III).

When we assessed the positivity rate of HRS cells reported by Gion et al., [25] we observed that 14/27 cases (51.9%) had scores of 4, followed by 8/27 (29.6%) with scores of 1, 1/27 (3.7%) with scores of 2, and 4/27 (14.8%) with scores of 3 (Table 3). Scores 1 and 2 represented the low-expression group, whereas scores 3 and 4 constituted the high-expression group. When comparing the groups with and without chemotherapy (Figure 4), the high PD-L1 expression group tended to have a higher rate of receiving chemotherapy. However, this difference was not statistically significant (*p* = 0.0808). The expression intensity and positivity rate in HRS cells were also determined and evaluated using modified H-scores, according to the methods of Roemer et al. (Table 4) [27], with a median value of 82.36 (range, 0.24–276.48). Furthermore, the two cases that showed Epstein–Barr virus (EBV) latency type I had lower H-scores than the median. There was no significant difference in the H-score between EBV-positive and EBV-negative CHL-IDD (Appendix A). In addition, we could not detect a correlation between the H-score and EBV latency (Appendix A).

### 3.4. Genetic Alterations to 9p24.1

The results of our FISH analysis are shown in Figure 5A–C. All patients exhibited CNAs in 9p24.1, with 16 (59%) classified as amplifications and 11 (41%) as copy gains. HRS cells included not only those with a CNA (amplification or copy gain) of 9p24.1 but also those exhibiting polysomy or disomy (Figure 5A).

Among the cases in the “amplification” group, 7–65% of the HRS cells showed copy number gains, 2–50% of the HRS cells showed 9p polysomy, and 4–88% of the HRS cells showed disomy. Similarly, “copy gain” cases contained 6–48% HRS cells showing polysomy and 25–80% HRS cells showing disomy (Figure 5A).

As is shown in Figure 5A, the proportion of CNAs varied in each case. However, the percentage of remaining disomic cells was notably higher in the cases with copy gains than in those with amplification (*p* = 0.0190; Figure 5B). Comparing the H-scores for PD-L1 with the proportion of remaining disomic cells revealed a significant inverse correlation (*p* = 0.0016; Figure 5C) that suggested a relationship between 9p24.1 CNAs and H-scores. No inverse correlation was revealed comparing PD-L1 H-score with each copy gain and amplification (*p* = 0.2264; Appendix A). Furthermore, we analyzed the relationship between the H-score and PD-L1 CNAs in EBV-positive and -negative cases of CHL-IDD. Although there was no inverse correlation in the EBV-negative CHL-IDD cases (*p* = 0.1363; Appendix A), we observed a trend in EBV-positive CHL-IDD cases (*p* = 0.0042; Appendix A). 

### 3.5. Relationship between 9p24.1 Genetic Alterations and Clinicopathological Characteristics

The patients were divided into two groups: a copy-gain group and an amplification group. The clinicopathological characteristics of both groups were evaluated. While we found no significant differences in terms of age, sex, or Eastern Cooperative Oncology Group Performance Status (PS) between the two groups, more patients in the copy gain group had CS III or higher than those in the amplification group, and there were significantly more extranodal lesions—as is indicated in Table 2 (*p* = 0.0002). 

In the copy gain group, extranodal involvement was observed in seven patients, with four having lung involvement. These included one patient with lung involvement only, two patients with lung involvement along with bone or pleural effusion, and one patient with lung involvement in addition to liver and adrenal involvement. The remaining three cases had adrenal glands, spinal cord, and bone marrow lesions. There was no significant difference in overall survival between the copy gain and amplification groups (Appendix A, *p* = 0.1228). However, there is a trend toward worse prognostic values in the first five years in the copy-gain group compared with the amplification group.

Regarding the association with EBV, 13/13 patients in the amplification group exhibited latency type II, whereas 2/8 patients in the copy gain group showed latency type I—which lacks LMP expression (case numbers 1 and 7 in Table 1). The CS of the cases with latency type I were IV and II. Both patients experienced SD after MTX withdrawal and were treated with chemotherapy. Additionally, 5/10 (50.0%) experienced regression after MTX withdrawal among the patients in the copy gain group, whereas 11/16 patients (68.6%) in the amplification group also had regression. In the copy-gain group, seven patients (two with Relapse/Regrowth following regression and five without regression) were treated with chemotherapy (4 ABVD, 1 A-AVD, 1 C-MOPP, and 1 Rituximab). Of these, three (42.8%) patients survived and four died. Among the amplification group, chemotherapy was required in 14 patients (9 with relapse/regrowth after regression and 5 without regression), and 11 patients were treated with chemotherapy (6 ABVD, 3 A-AVD, 1 R-CHOP, and 1 CPA). Among them, nine survived (81.8%), and two died (Figure 3B). No significant differences in terms of responses following MTX withdrawal were observed between the amplification and copy gain groups. There were no significant differences in terms of the need for chemotherapy (Appendix A). The median time to chemotherapy following MTX withdrawal was eight months in the copy-gain group and nine months in the amplification group. 

## 4. Discussion

CHL-IDD and de novo CHL exhibit morphological similarities and are difficult to differentiate without detailed clinical information. However, they differ in clinical course, treatment approaches, and clinical outcomes [13,25,29]. In this study, we assessed 9p24.1 CNAs and PD-L1 expression using FISH and immunohistochemistry to elucidate their relationships with clinical features.

Roemer et al. reported a correlation between the modified H-score of PD-L1 protein expression and 9p24.1 CNAs in cases of de novo CHL [27]. Our study also observed a significant correlation between PD-L1 protein expression and 9p24.1 CNAs, as determined by modified H-score CNAs in the CHL-IDD (*p* = 0.0016). This result is consistent with the findings of Shiraiwa et al. [28].

We also observed disomic cells in tumor cells in all cases analyzed in this study. Thus, both CHL-IDD and de novo CHL tumor cells exhibit heterogeneity in the genomic aberration of the 9p24.1 region containing the *CD274* gene that encodes PD-L1 within the tumor. These results indicate similar genomic events were provoked in CHL-IDD and de novo CHL.

Concerning PD-L1 protein expression, one report suggested that the time until the start of chemotherapy was notably shorter in those patients with CHL-IDD with higher levels of PD-L1 expression [25]. In that study, patients with high levels of PD-L1 expression required chemotherapy more frequently than the low-expression group. Though we did not detect significant differences in terms of the time until chemotherapy initiation, we observed a trend toward recurrence or an increase in the number of patients requiring chemotherapy in the high-expression group compared to the low-expression group. Therefore, the expression level of PD-L1 may correlate with CHL-IDD progression, similar to de novo CHL.

We also investigated the association between 9p24.1 and the clinicopathological features of CHL-IDD. In cases of de novo CHL, the presence of 9p24.1 CNAs has been observed in nearly all instances [27,30], and it has been demonstrated that patients with amplification have a significantly shorter PFS [27]. Shiraiwa et al. examined 7 cases of CHL-IDD and observed amplification in 6/7, while copy gain was found in only 1 case [28]. Although the sample size was small, it was suggested that amplification of 9p24.1 is a common event in CHL-IDD. In this study, we also observed that all patients with CHL-IDD exhibited CNAs, consistent with the findings of a previous report [28]. We observed 11 cases of copy gain and 16 cases of amplification. Though no significant relationship was observed between abnormalities in the 9p24.1 region and the clinical outcomes (PFS and overall survival [OS]) in CHL-IDD, there was a trend toward worse prognostic values in the first five years in the copy-gain group compared with the amplification group. Furthermore, PD-L1 expression did not significantly affect PFS or OS. These observations suggest that, in the context of de novo CHL, the expression of PD-L1 may play a role in immune evasion. However, it is possible that underlying conditions, such as the immunosuppressive state during MTX administration, and other comorbidities, such as RA, may also influence the clinical outcomes of CHL-IDD.

While previous reports have indicated that amplification was more frequently observed in the higher clinical stages (CS III/IV) of de novo CHL [27], in this study, the copy-gain group of patients with CHL-IDD had a higher incidence of CS III/IV than the amplification group, and the copy gain group frequently presented with extranodal lesions. These findings suggest that, in contrast to de novo CHL, CHL-IDD may involve pathological mechanisms that depend on other mechanisms, along with the amplification of 9p24.1, which contains the *CD274* region that encodes PD-L1. This phenomenon may represent a unique characteristic specific to CHL-IDD. Within the copy-gain group of our cohort, 2/8 cases exhibited EBV latency type I, which contrasted with the absence of latency type I in the amplification group. The EBV latency pattern in the copy gain group also differed from the typical latent infection mode observed in cases of de novo CHL. Furthermore, the presence of extranodal lesions in 7/10 cases, specifically within the copy gain group, indicates that this unique condition may be affected by factors other than 9p24.1 amplification in some cases of CHL-IDD with copy number gains. CHL is characterized by a low absolute number of HRS cells and various inflammatory cells in the surrounding immune microenvironment. In this context, the expression of PD-L1 in HRS cells plays a crucial role in immune evasion by PD-1-positive cytotoxic T lymphocytes and serves as a central mechanism in the pathogenesis of de novo CHL. Higher PD-L1 expression in de novo CHL has significantly contributed to tumor immune evasion. Before the introduction of PD-1 antibody therapy, standard treatments were less effective, and higher PD-L1 expression was associated with resistance and higher recurrence rates to standard chemotherapy [31]. However, we observed unexpected results in more cases in the copy-gain group that present low PD-L1 expression with clinically aggressive cases of CHL-IDD. This result may be attributed to another immune checkpoint pathway operating independently of the PD-L1/PD-1 axis. Nevertheless, further investigation is required to explore this possibility.

Roemer et al. reported higher PD-L1 H-scores in cases of EBV-positive CHL, suggesting that EBV infection further induces PD-L1 expression [27]. It was reported that PD-L1 ligand expression was upregulated by EBV infection in cases of EBV-positive de novo CHL [32] and it has also been shown that LMP1 plays a significant role in PD-L1 expression [33,34]—not only in EBV-positive LPD but also in other cancers such as nasopharyngeal, gastric, and breast cancers [35,36,37,38]. Our results in this study showed a correlation between PD-L1 H-score and PD-L1 CNAs in EBV-positive CHL-IDD cases, but we did not observe a significant difference in PD-L1 expression in EBV-negative cases (Appendix A). However, it is worth noting that the frequency of EBV positivity in CHL-IDD is higher than in de novo CHL, as reported previously [29]. Although a relationship between EBV infection and PD-L1 expression was observed in our cases of de novo CHL, we did not find a significant relationship between them in cases of CHL-IDD. Therefore, a more complicated mechanism of PD-L1 expression may underlie CHL-IDD.

## 5. Conclusions

Our study confirmed the presence of PD-L1 protein expression and 9p24.1 CNAs in CHL-IDD, exhibiting patterns similar to those in de novo CHL. Notably, within the copy gain group, extranodal lesions that were not observed in the amplification group were identified in approximately two-thirds of the cases. The high CS in these cases suggests that these lesions represent a distinctive subtype within CHL-IDD. Further research is required to clarify a comprehensive understanding of the distinct clinical pathophysiology of CHL-IDD.

## Figures and Tables

**Figure 1 cancers-16-01298-f001:**
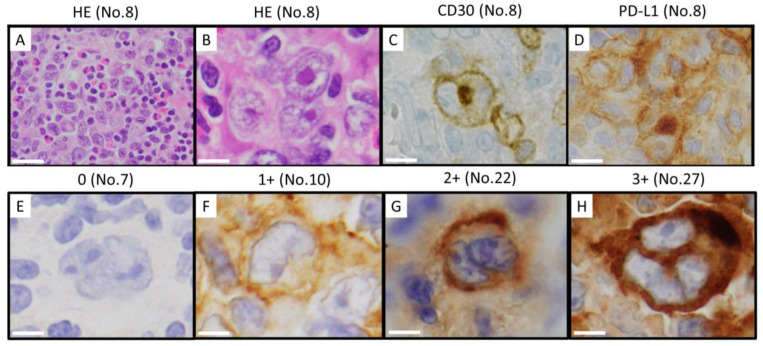
Morphological definition of CHL-IDD. Evaluation of programmed cell death-ligand 1 (PD-L1) positive scores in classic Hodgkin lymphoma arising in immune deficiency/dysregulation (CHL-IDD). Each number indicates the case number in Table 1. Hematoxylin and eosin staining at low-power (**A**) and high-power (**B**) fields. Hodgkin/Reed-Sternberg (HRS) cells were observed in cellular backgrounds that were rich in lymphocytes and histiocytes. HRS cells were CD30-positive (**C**) and PD-L1-positive (**D**). No. 27 showed an amplification of 9p24.1, with an H-score of 276.5. No. 8 showed copy gain, with an H-score of 82.36. Staining intensity in tumor cells is classified as 0: absent (**E**), 1+: weak (**F**), 2+: moderate (**G**), and 3+: strong (**H**). Scale bars, 50 μm (**A**), 20 μm (**B**–**H**).

**Figure 2 cancers-16-01298-f002:**
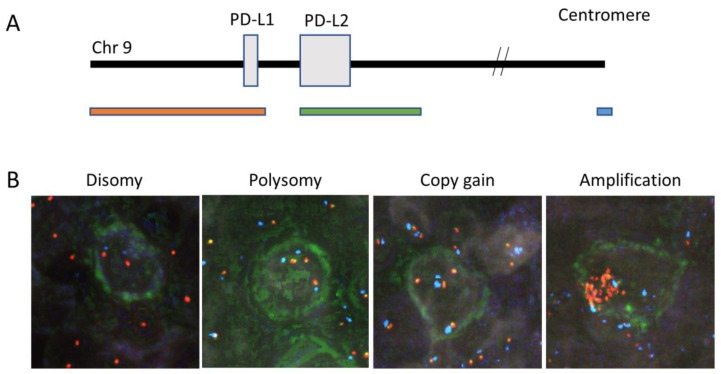
A location of probe and representative images of fluorescence in situ hybridization results for the various categories. (**A**) Location of each probe used in the study. RP11-599H20, including PD-L1, was labeled in orange. RP11-635N21, including PD-L2, was labeled in green and the centromeric probe (CEP9) is in aqua. (**B**) Representative images of fluorescence in situ hybridization results for the various categories. In these images, disomy reflects 2Aqua (2A): 2 Fusion (2F) (case no.24); polysomy, 3A: 3F (case no.27); copy gain, 2A: 4F (case no.11); and amplification, 3A: 15F or over (case no.27).

**Figure 3 cancers-16-01298-f003:**
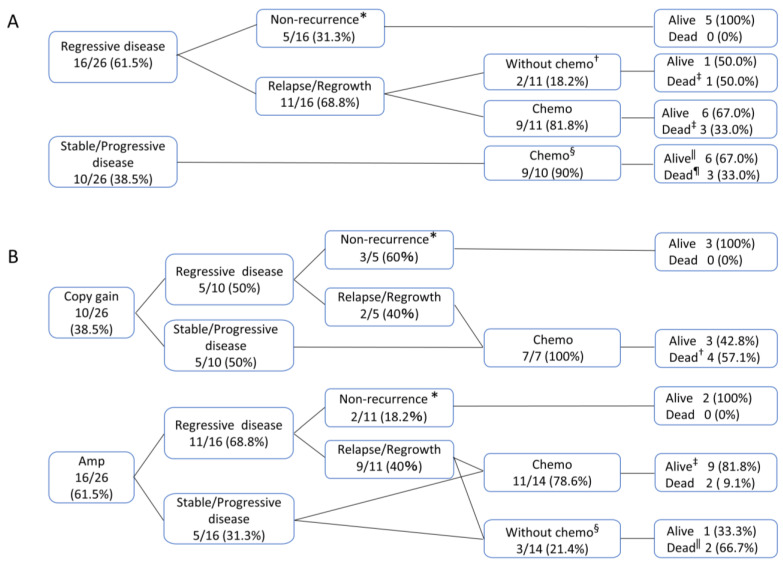
The clinical course of patients with CHL-IDD. (**A**) The clinical course of patients with classic Hodgkin lymphoma arising in immune deficiency/dysregulation (CHL-IDD) after methotrexate (MTX) withdrawal. * Includes two cases with a short observation period due to transfer. † One patient had a severe condition that could not be treated, while the other patient was treated with prednisolone (PSL) alone. ‡ Death due to current disease. § One patient refused treatment. ‖ One patient’s details were unknown, and one patient had a short observation period. ¶ Dead includes two patients who died due to their current diseases and one who developed primary biliary cirrhosis and died as a result of disseminated intravascular coagulation. (**B**) The clinical course of patients with CHL-IDD after MTX withdrawal (copy gain and amplification). * Includes one case with a short observation period due to transfer. † Dead includes three patients who died due to their current diseases and one who developed primary biliary cirrhosis and died as a result of disseminated intravascular coagulation. ‡ One patient’s details were unknown, and one patient had a short observation period. § One patient had a severe condition that could not be treated, one patient refused treatment, and one was treated with PSL alone. ‖ Death of current disease. Chemo, chemotherapy.

**Figure 4 cancers-16-01298-f004:**
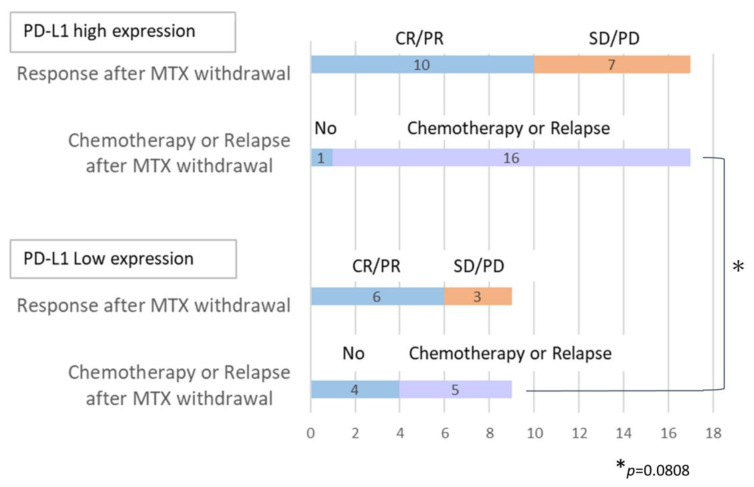
Changes in disease condition after MTX withdrawal in patients with CHL-IDD. Changes in lesions following methotrexate (MTX) withdrawal in tumor cells with high PD-L1 expression and low PD-L1 expression.

**Figure 5 cancers-16-01298-f005:**
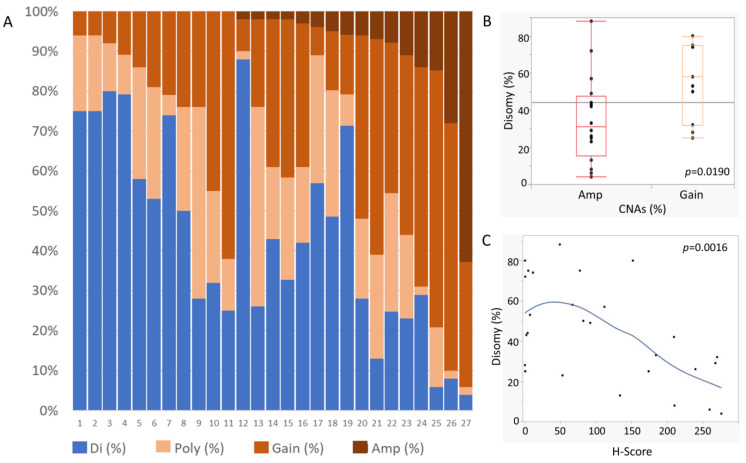
A spectrum of 9p24.1 copy number alterations in patients with CHL-IDD. (**A**) A column represents each CHL-IDD sample. In each sample, the percentage of Hodgkin/Reed-Sternberg (HRS) cells with 9p24.1 disomy (blue), polysomy (light pink), copy gain (pink), and/or amplification (dark pink) is shown on the *y*-axis. (**B**) Percentage of disomic HRS cells grouped by copy number alteration category. (**C**) The association of PD-L1 expression and 9p24.1 CNAs. Residual 9p24.1 disomy is depicted on the *y*-axis. H-score is shown on the *x*-axis. Di, disomy; Poly, polysomy; Gain, copy gain; Amp, amplification.

**Table 1 cancers-16-01298-t001:** Clinical findings of patients with CHL-IDD.

Pt. No	Age/ Sex	Primary Basal Disease	Anti-RA Drugs at the Time of LPD Development	Clinical Stage	Extranodal Involvement Site	EBER/ Latency Pattern	Response after MTX Withdrawal	Relapse after MTX Withdrawal	Time to Chemotherapy (Month)	Chemotherapy	Duration after Dx (Year)	Alive/ Dead	Cause of Death
1	67/F	RA	MTX	II	-	I	SD	+	2.8	ABVD	1.92	Dead	DIC, PBC
2	70/M	RA	MTX	IV	lung, bone	II	PR	-	-	-	2.69	Alive	
3	67/M	RA	MTX, PSL, Sodium Aurothiomalate	NR	-	II	CR	-	-	-	2.34	Alive	
4	45/F	RA	MTX	IV	bone marrow	II	PD	-	0.1	Rituximab, ADR, CP	0.24	Dead	PD
5	68/F	RA	MTX, PSL	IV	adrenal gland	II	PR	NR	-	-	0.18	Alive	
6	77/F	RA	MTX, PSL, BUC	IV	lung, liver, adrenal gland	EBER-	SD	+	44.0	A-AVD	4.00	Alive	
7	43/M	RA	MTX	IV	lung	I	SD	+	1.0	ABVD	4.83	Alive	
8	79/F	RA	MTX, SSZ, PSL	III		II	NR	NR	NR	A-AVD	0.25	Dead	Unk
9	71/F	RA	MTX, PSL, SSZ, TAC	IV	lung, pleural effusion	II	SD	+	8.0	C-MOPP	0.25	Dead	PD
10	69/F	RA	SASP, PSL, MTX, BUC	III	-	EBER-	CR	+	9.0	ABVD	0.58	Dead	PD
11	59/F	RA	MTX, TAC, ETN, Abatacept, TCZ, SASP, PSL, peficitinib	IV	paravertebral	EBER-	PR	+	34.0	ABVD	9.75	Alive	
12	41/M	SAPHO synd.	MTX, PSL, INF	I	-	II	CR	+	17.0	ABVD	5.96	Alive	
13	70/F	RA	MTX, TAC, ETN, ADA, TCZ	III	-	II	SD	+	16.0	ABVD	4.50	Alive	
14	57/F	RA	MTX, CyA, TAC, INF, ADA, TCZ	I	-	EBER-	CR	+	9.0	ABVD	6.42	Alive	
15	63/F	RA	MTX, PSL	III	-	II	CR	+	1.4	ABVD	2.07	Alive	
16	67/M	RA	MTX, MZB, SSZ	III	-	II	PR	+	23.0	A-AVD	3.17	Alive	
17	64/M	RA	MTX, PSL, BUC	III	-	II	CR	+	8.0	ABVD	5.17	Dead	PD
18	48/F	RA	MTX, PSL, TAC	III	-	EBER-	SD	NR	1.2	ABVD	9.03	Alive	
19	65/M	RA	MTX	I	-	II	PR	-	-	-	14.08	Alive	
20	68/F	RA	MTX, PSL, TAC	II	-	II	PD	-	27.8	ABVD	9.50	Alive	
21	82/F	RA	MTX	III	-	II	CR	-	-	-	1.07	Alive	
22	79/F	RA	MTX	III	-	II	PR	+	7.6	ABVD	1.84	Alive	
23	61/F	RA	MTX,	I	-	II	PR	+	25.0	CP	2.16	Dead	PD
24	57/M	RA	MTX, PSL	II	-	ND	PD	-	2.3	R-CHOP	1.02	Alive	
25	81/F	RA	MTX, ETN, PSL	II	-	II	SD	+	-	-	0.58	Dead	PD
26	71/F	RA	MTX, PSL	III	-	II	PR	+	-	-	1.67	Alive	
27	79/M	RA	BUC, MTX, SSZ, TAC	III	-	II	PR	+	-	-	0.08	Dead	PD

A-AVD, brentuximab-vedotin + adriamycin + vinblastine + dacarbazine; ABVD, adriamycin + bleomycin + vinblastine + dacarbazine; ADA, adalimumab; ADR, adriamycin; BUC, bucillamine; C-MOPP, cyclophosphamide + procarbazine + vincristine + prednisone; CPA, cyclophosphamide; CR, complete remission; CyA, Cyclosporine; EBER, Epstein-Barr Virus-encoded small RNA; F, Etanercept, ETN; female; INF, infliximab; M, male; MTX, methotrexate; MZB, mizoribine; ND, not done; NR, not recorded; PBC, Primary biliary cirrhosis; PD, progressive disease; PR, partial remission; PSL, prednisolone; RA, rheumatoid arthritis; R-CHOP, Rituximab + Cyclophosphamide + hydroxydaunorubicin + oncovin+ prednisone; RD, regressive disease; SD, stable disease; SSZ, sulfasalazine; TAC, tacrolimus; Tocilizumab, TCZ; Unk, unknown.

**Table 2 cancers-16-01298-t002:** Clinical data of the copy gain group and amplification group were categorized by 9p24.1 CNAs.

		ALL	Copy Gain	Amp	*p*-Value
	N	27	11	16	
Clinical features	Age (range)	67 (41–82)	68 (43–79)	66 (41–82)	0.9516
	Sex (M/F)	9/18	3/8	6/10	0.6828
	LDH > normal upper range	12/25 (148–442)	4/10 (196–390)	8/15 (148–442)	0.9902
	sIL2R > normal upper range	21/23 (409–4981)	8/10 (409–4000)	13/13 (486–4981)	0.4833
	PS ≥ 2	15.0% (3/20)	25.0% (2/8)	8.3% (1/12)	0.5368
	Clinical stage ≥ 3	69.2% (18/26)	90.0% (9/10)	56.3% (9/16)	0.0989
	Clinical stage 3	42.3% (11/26)	20.0% (2/10)	56.3% (9/16)	0.0687
	Ratio of extranodal Involvement	26.9% (7/26)	70% (7/10)	0% (0/16)	0.0002
	B symptom positive	71.4% (15/21)	88.9% (8/9)	58.3% (7/12)	0.1778
	EBER positive	80.8% (21/26)	72.7% (8/11)	86.7% (13/15)	0.6196
	MTX	100% (27/27)	100% (11/11)	100% (16/16)	1
Anti-RA drugs at the time of LPD development	Prednisolone	55.6% (15/27)	63.6% (7/11)	50.0% (8/16)	0.696
	Tacrolimus	25.9% (7/27)	18.2% (2/11)	31.3% (5/16)	0.6618
	Other immunosuppressive drugs	51.9% (14/27)	54.5% (6/11)	50.0% (8/16)	1
	Regression	61.5% (16/26)	50.0% (5/10)	68.8% (11/16)	0.425
Clinical response after MTX discontinuation	Ratio	69.2% (18/26)	70.0% (7/10)	68.8% (11/16)	1
Chemotherapy after MTX-LPD	Response: CR/PR	81.3% (13/16)	71.4% (5/7)	88.9% (8/9)	0.55
	Alive/Dead	18/9	6/5	12/4	0.4105

CR/PR, Complete remission/partial remission; CNAs, copy number alteration; EBER, Epstein–Barr Virus-encoded small RNA; LDH, lactate dehydrogenase; LPD, lymphoproliferative disorder; M/F, male/female; MTX, methotrexate; PS, performance status; RA, rheumatoid arthritis; sIL2R, soluble Interleukin 2-receptor.

**Table 3 cancers-16-01298-t003:** Immunohistochemical and EBV latency date of copy gain group and amplification group categorized by 9p24.1 CNAs.

		ALL	Copy Gain	Amp	*p*-Value
N		27	11	16	
CD30		100% (27/27)	100% (11/11)	100% (16/16)	1
CD15		72.7% (16/22)	50.0% (4/8)	85.7% (12/14)	0.1365
CD20		37.0% (10/27)	36.4% (4/11)	37.5% (6/16)	1
CD79a		37.0% (10/27)	45.5% (5/11)	31.3% (5/16)	0.6868
PD-L1		100% (27/27)	100% (11/11)	100% (16/16)	1
PD-L1	Score 1	29.6% (8/27)	36.4% (4/11)	25% (4/16)	0.5873
	Score 2	3.7% (1/27)	9.1% (1/11)	0% (0/16)	
	Score 3	14.8% (4/27)	9.1% (1/11)	18.8% (3/16)	
	Score 4	51.9% (14/27)	45.5% (5/11)	56.3% (9/16)	
EBER-ISH		81.5% (22/27)	72.7% (8/11)	87.5% (14/16)	0.3705
LMP-1		73.1% (19/26)	54.5% (6/11)	77.8% (13/15)	0.0946
EBNA2		0.00% (0/15)	0.00% (0/7)	0.00% (0/8)	1
Latency	type I	9.5% (2/21)	25% (2/8)	0% (0/13)	0.1333
	type II	90.5% (19/21)	75% (6/8)	100% (13/13)	
	type III	-	-	-	

Amp, amplification; CNAs, copy number alterations; EBER, Epstein–Barr Virus-encoded small RNA; EBNA2, Epstein–Barr virus nuclear antigen 2; EBV, Epstein–Barr virus; PD-L1, programmed cell death-ligand 1.

**Table 4 cancers-16-01298-t004:** H-score of PD-L1 expression by immunohistochemical studies in 27 patients with CHL-IDD.

Case No.	H-Score	Ave. Score	% of PD-L1positive HRS Cells
9	0.24	0.06	4
3	0.36	0.06	6
19	0.64	0.08	8
21	0.64	0.08	8
14	2.20	0.22	10
23	4.00	0.33	12
2	4.88	0.33	15
6	7.28	0.28	26
7	11.52	0.48	24
12	49.28	0.88	56
22	53.04	1.02	52
5	67.00	0.85	79
1	77.52	1.02	76
8	82.36	1.42	58
18	92.01	1.28	72
17	112.00	1.40	80
20	133.92	1.86	72
4	152.00	1.90	80
11	174.18	2.08	84
15	184.52	1.91	96
16	209.84	2.44	86
26	210.60	2.34	90
13	240.00	2.40	100
25	260.00	2.60	100
24	268.00	2.68	100
10	270.48	2.76	98
27	276.48	2.88	96

The average (Ave.) score was calculated by multiplying each staining intensity (1 to 3+) by the percentage of HRS cells with positive staining (0% to 100%) and dividing by the total number of PD-L1-positive HRS cells. The H-score is calculated by multiplying the Ave. score by % of positive HRS cells. MTX, methotrexate; PD-L1, programmed cell death-ligand 1.

## Data Availability

All data supporting the findings of this study are available within the paper and its Appendix A.

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
