# Peer review of "Copy Number Analysis of 9p24.1 in Classic Hodgkin Lymphoma Arising in Immune Deficiency/Dysregulation"

_cancers, 2024, doi:10.3390/cancers16071298_

Round 1

Reviewer 1 Report

Comments and Suggestions for Authors

Ohsawa et al., in Copy Number Analysis of 9p24.1 in Classic Hodgkin Lymphoma Arising in Immune Deficiency/Dysregulation

Overall, the study is an excellent addition to the literature on the incidence of 9p24.1 alterations in the Hodgkin RS cells of CHL and the associations between EBV status, magnitude of 9p24.1 gains, and PD-L1 protein expression for immuno-deficiency associated CHL.  This work builds upon the prior work in this area, which had primarily focused on de novo CHL rather than immuno-deficiency-associated CHL.  

The authors are commended for their detailed clinical information and treatment and clinical outcomes summary (i.e., figure 3), which is quite informative. The paper is well written, the methods appropriate, and the conclusions sound. 

There are only minor suggestions to the manuscript:

1.    Line 33, is PD-L1 protein expression in Hodgkin cells or the tumor microenvironment (i.e. macrophages) or both?

2.    Line 76, Roemer et al used fluorescence in situ hybridization (FISH) studies to detect 9p24.1 alterations not FICTION.

3.    In Table 4- what is the “average score” and how does this differ from the H-score?  It might be good to explain this in the legend for the table. And for the same table, what does “% of positive cells” mean? This is the percentage of all cells positive for PD-L1 or the percentage of all Hodgkin cells positive for PD-L1?

4.    Line 283—what is CS3?  Presumably, this is “AJCC clinical stage III”?

5.    Line 302—wrote “Four4” as a typo.

6.    Line 368—do the authors mean PD-1-positive cytotoxic T lymphocytes instead of PD-L1-positive cytotoxic T lymphocytes?

7.    Did any of the patients, at any time, receive anti-PD-1?

Reviewer 2 Report

Comments and Suggestions for Authors

Ohsawa et al. analyzed HRS cell 9p24.1 copy numbers and PD-L1 expression in classical Hodgkin lymphoma (cHL) that developed in rheumatoid arthritis (RA) patients on immunosuppressive therapy and described subgroups in relation to their PD-L1 expression and chromosome 9p24.1 copy number alterations in this paper. They also analyzed EBV status of HRS cells and clinical outcome in relation to these two characteristics. The observation of increased extranodal disease in the “gain” group is interesting and the outcome in each group and in the entire study population provides useful data. EBV data is also of interest; however, the numbers are too small to conclude. Similarly, drawing major conclusions from the entire study is difficult. They report on a total of 27 cases. Since cHL as a complication of immunosuppressive therapy in RA is not that common, the number included in this report is sizeable and the observations may serve as a reference for larger scale studies. 

Major 

1.     The authors analyzed the cases from different perspectives and tried to compare all these variables with each other in and between groups. Though 27 cases is a decent number for this type of a report as mentioned before, drawing conclusions, particularly, based on the observations is not feasible and leads to many speculations. 

2.     Several results given in the text are also available the tables; thus, repeating the same data in the text causes unnecessary extension of the manuscript and confusion for the reader. Some essential/critical aspects can be restressed in the text but observations on small number of cases due to their certain characteristics should be avoided. It is not flowing smoothly for the reader. Text must be shorten significantly. 

3.      There are several repeats in results and discussion sections of the text again leading to difficulty in understanding and gets confusing at times. 

4.     As stated in the report, there is an apparent increase in extranodal cases in the “gain” group and this is interesting. However, the authors lump the clinical stage (CS) III and IV (assuming that all with extranodal cases did not have nodal direct extension to the lungs, e.g. not stage IIE) in their comparison. Thus, if only CS III cases included, “gain” group had 2 and “amplification” group 7. If an analysis were run in CS III cases, there may be a statistical significance between the groups. 

5.     The authors need to clarify the definition of “gain” and “amp” groups based on H-score for the readers who are not familiar with this approach. It is demonstrated that all the samples on 27 cases had some percent of “gain” (Figure 5). There is a need to clarify in the text that if even a small percent of “amp” observed in a given sample, that case is classified as an “amp”. 

6.     Based on the results (Figure 5), both groups have varying percent of disomies. The authors compared the two groups for percent disomies and found an increased disomy percent in the “gain” group. Then compared disomy content and PD-L1 expression (H-score) and observed and inverse relationship between disomy content and PD-L1 expression. Then extrapolated this data to conclude PD-L1 expression is higher in the “amp” group; this is an indirect comparison. It makes me wonder then why “gain” or “amp”  status would be important. Though, it was stated in the discussion section that some others used the same approach, why not compare H-scores between “gain” and “amp” group directly and see if there is any difference? 

7.     Although it is not the scope of this report, the results would have been more meaningful if the cHL-IDD case results were compared with institutional de novo cHL patient results. 

Minor

1.     It is somewhat difficult to recognize PD-L2 probe-green fluorescence labels in Figure 2. Using arrowheads may help.

2.     Occasional typos: not quite 2025, yet (line #89); x2 fours (line #302); missing parenthesis (line #345)

Comments on the Quality of English Language

Minor edits are necessary.

Round 2

Reviewer 2 Report

Comments and Suggestions for Authors

The authors have responded somewhat to the raised issues. This paper will contribute to medical literature due to certain interesting observations. 

Author Response

Dear Reviewer 2

Thank you for your comment.

We will continue performing research according to your suggestions.